

# Unimodal head-width distribution of the European eel (*Anguilla anguilla* L.) from the Zeeschelde does not support disruptive selection

Pieterjan Verhelst[1,2,3,4], Jens De Meyer[5], Jan Reubens[4], Johan Coeck[2], Peter Goethals[3], Tom Moens[1] and Ans Mouton[2]

[1] Marine Biology Research Group, Ghent University, Ghent, Belgium
[2] Research Institute for Nature and Forest, Brussels, Belgium
[3] Laboratory of Environmental Toxicology and Aquatic Ecology, Ghent University, Ghent, Belgium
[4] Flanders Marine Institute, Ostend, Belgium
[5] Evolutionary Morphology of Vertebrates, Ghent University, Ghent, Belgium

Corresponding author
Pieterjan Verhelst,
Pieterjan.Verhelst@UGent.be

## ABSTRACT

Since the early 20th century, European eels (*Anguilla anguilla* L.) have been dichotomously classified into 'narrow' and 'broad' heads. These morphs are mainly considered the result of a differential food choice, with narrow heads feeding primarily on small/soft prey and broad heads on large/hard prey. Yet, such a classification implies that head-width variation follows a bimodal distribution, leading to the assumption of disruptive selection. We investigated the head morphology of 272 eels, caught over three consecutive years (2015–2017) at a single location in the Zeeschelde (Belgium). Based on our results, BIC favored a unimodal distribution, while AIC provided equal support for a unimodal and a bimodal distribution. Notably, visualization of the distributions revealed a strong overlap between the two normal distributions under the bimodal model, likely explaining the ambiguity under AIC. Consequently, it is more likely that head-width variation followed a unimodal distribution, indicating there are no disruptive selection pressures for bimodality in the Zeeschelde. As such, eels could not be divided in two distinct head-width groups. Instead, their head widths showed a continuum of narrow to broad with a normal distribution. This pattern was consistent across all maturation stages studied here.

# INTRODUCTION

*Törlitz*'s (*1922*) introduction of the terms 'narrow' and 'broad' headed eels (genus *Anguilla*) led to numerous studies trying to explain these two distinct morphs. Eels are highly flexible species with a complex life cycle. They develop as leptocephalus larvae into glass eels in the oceans, and settle as elvers in coastal and/or freshwater habitats where they grow during what is commonly known as the yellow eel stage. When eels have reached a threshold size

and physiological condition, including sufficient fat reserves, they migrate back to their spawning site as silver eels (*Tesch, 2003*).

A plausible explanation for the head dimorphism is disruptive selection via resource polymorphism resulting in phenotypic plasticity, a phenomenon that occurs in many vertebrates, especially fish (*Skulason & Smith, 1995*), and that essentially enables individuals of the same species to reduce intraspecific competition through resource selectivity (*Schoener, 1974*; *Svanbäck et al., 2008*). Differences in consumed prey, for example, can lead to morphological variation in the feeding apparatus. Such a relation between feeding ecology and morphology of the feeding apparatus has been well established in animals (*Iijima, 2017*; *Muschick et al., 2011*; *Saunders & Barclay, 1992*). A similar relation between feeding ecology and morphology has been observed in both the European (*Anguilla anguilla* L.) and Japanese eel (*A. japonica* Temminck & Schlegel). Several studies have illustrated a link between feeding strategy and head width, with narrow headed eels feeding on small and/or soft prey (e.g., amphipods and chironomids) and broad headed eels on large and/or hard prey (e.g., molluscs and fish) (*Cucherousset et al., 2011*; *De Meyer, Christiaens & Adriaens, 2016*; *Kaifu et al., 2013*; *Lammens & Visser, 1989*; *Micheler, 1967*; *Proman & Reynolds, 2000*). The broader heads thus reflect better developed jaw closing muscles and a relatively broader skull, features which facilitate the consumption of hard and/or large prey items (*De Meyer, Christiaens & Adriaens, 2016*).

Yet, the European eel is an opportunistic animal (*Lammens et al., 1985*; *Schulze et al., 2004*; *Van Liefferinge et al., 2012*), although specialization on specific prey items has been observed (*Barak & Mason, 1992*), challenging the dichotomous and strongly deterministic characterization into 'broad' and 'narrow' heads based on feeding behavior. Indeed, head dimorphism may not be entirely attributed to differences in foraging. For instance, narrow headed Japanese eels grow faster than broad heads (*Kaifu et al., 2013*) and genetic support for this hypothesis has recently been found in European eel (*De Meyer et al., 2017b*). Moreover, certain genes involved in growth speed, such as *growth hormone-1*, are also involved in salinity preference (*Iwata et al., 1990*); thus, eels preferring freshwater grow more slowly than eels favoring marine waters (*Edeline, Dufour & Elie, 2005*). Hence, the basis for head dimorphism in eels may be much more complex than originally thought.

Despite substantial research related to eel head widths, many knowledge gaps remain. For instance, head width of glass eels follows a unimodal distribution (*De Meyer et al., 2015*). Consequently, a strict dichotomous division of such glass eels into a narrow and a broad headed morph is impossible, as a gradual transition exists from narrower to broader headed eels with many intermediate forms. Still, many studies have dichotomously classified narrow and broad headed eels using a ratio-based threshold: eels with a head width over total body length ratio smaller than 0.033 are considered narrow heads, while eels with larger ratios are broad heads (*Barry et al., 2016*; *Kaifu et al., 2013*; *Lammens & Visser, 1989*; *Proman & Reynolds, 2000*). However, head width increases allometrically with total length (*De Meyer et al., 2017a*; *De Meyer et al., 2015*; *Lammens & Visser, 1989*), so larger eels may be wrongly classified as broad heads.

In contrast to the above-mentioned unimodal head-width distribution in glass eels, the head width of yellow eels has been suggested to follow a bimodal distribution (*Ide et al.,*

*2011*; *Kaifu et al., 2013*). Bimodality would occur during the maturation stage after glass eel settlement. Six different maturation stages have been identified from the yellow eel stage onwards (*Durif, Dufour & Elie, 2005*): a sexually undifferentiated yellow stage (I), a female yellow stage (FII), a female intermediate stage (FIII), two female silver eel stages (FIV and FV) and a male silver eel stage (MII). It is therefore possible that the unimodality found in glass eels shifts to bimodality during further development through these stages.

From an evolutionary point of view, variations in head shape may arise from different selective pressures at many locations, or even disruptive pressures such as observed on a side channel of the Frome River (*Cucherousset et al., 2011*): individuals with intermediate traits would have a lesser fitness than individuals with more extreme traits, because they may be less efficient in the consumption of both soft/small prey and hard/large prey in comparison to the more extreme morphs (*Martin & Pfennig, 2009*). Head morphology may also affect an eel's fitness in yet another way: narrow-headed eels have a more hydrodynamic body shape and may therefore migrate faster or in a more energetically favorable way than broad heads (*De Meyer, Christiaens & Adriaens, 2016*; *Van Wassenbergh, Potes & Adriaens, 2015*), increasing their chances of successful spawning.

In this study, we hypothesize that eels from a single river drainage do not show disruptive selection related to eel head width by assessing four sub-hypotheses: (1) head-width variation follows a unimodal distribution, and (2) this distribution does not differ between different maturation stages; (3) body condition does not differ according to head width, and (4) eels with a narrower head width migrate at a similar speed as eels with a broader head width.

## METHODS

### Study area

The River Schelde is approximately 360 km long and has a drainage area of 21,863 km$^2$ (Fig. 1). The river originates on the plateau of Saint-Quentin in France and runs through Belgium into the North Sea in The Netherlands. The Schelde is one of the few European rivers with a well-developed estuary. It is approximately 160 km long and has a complete salinity gradient from marine to a tidal freshwater zone, including extensive freshwater, brackish and salt marshes. The Belgian part of the Schelde Estuary (i.e., the Zeeschelde) runs from Gent to Antwerp. It is well-mixed and characterized by strong currents, high turbidity and a large tidal amplitude up to 6 m (*Seys, Vincx & Meire, 1999*). It has a length of 105 km, a width of 50 m to 1,350 m, and an average discharge of 100 m$^3$ s$^{-1}$. In addition, several tributaries discharge into the Zeeschelde. Our study area only comprised the Zeeschelde. There is no commercial fishing in this area and fyke fishing is prohibited in Belgium since 2009, yet, recreational fishing for eels does occur.

### Data collection

Over three consecutive years (i.e., 2015 till 2017), 272 eels were caught in summer and autumn with double fyke nets (mesh size = 8 mm) downstream the tidal weir (Merelbeke) in the freshwater part of the Zeeschelde. The dorsal view of the head was photographed with a digital camera on graph paper and several morphometric features were measured

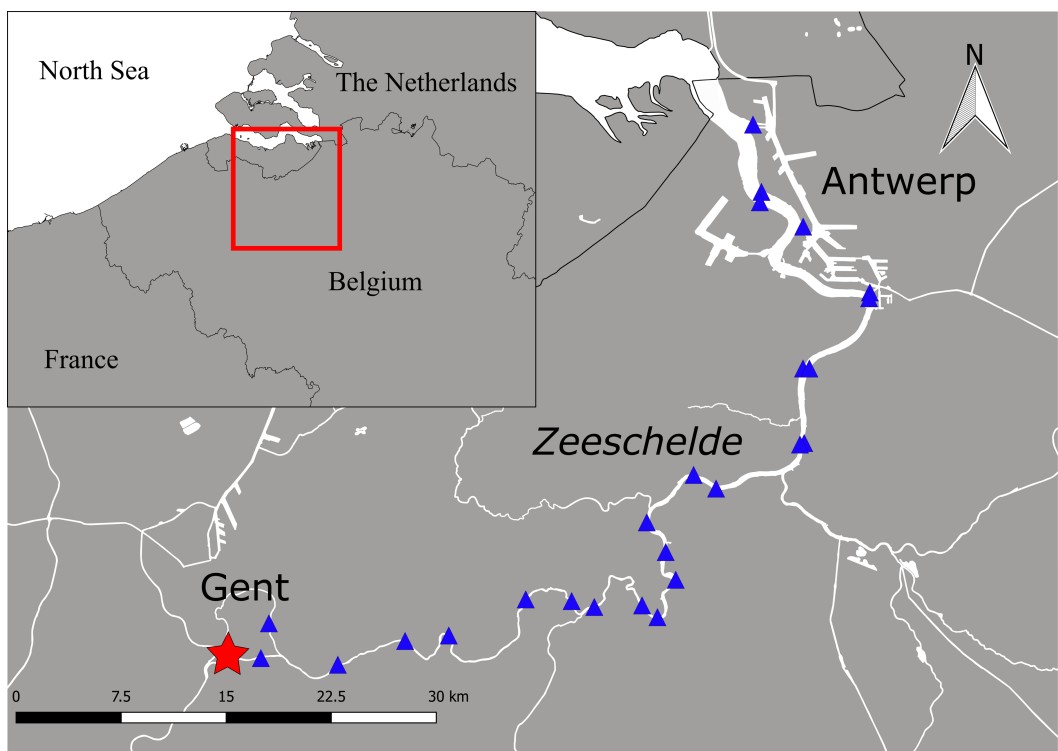

**Figure 1** Study area with the catch location at the tidal weir in Merelbeke (asterisk) and the position of the acoustic listening stations (triangles) in the Zeeschelde.

in order to determine the eel maturation stages according to *Durif, Dufour & Elie (2005)*: total length (TL, to the nearest mm), body weight (W, to the nearest g), the left vertical and horizontal eye diameter ($ED_v$ and $ED_h$ respectively, to the nearest 0.01 mm) and the length of the left pectoral fin (FL, to the nearest 0.01 mm) (Table 1). Eels of all six different maturation stages were caught: sexually undifferentiated yellow eels (I, $n = 51$), female yellow eels (FII, $n = 68$), premigrant female eels (FIII, $n = 91$), two female silver eel stages (FIV and FV, $n = 15$ and $n = 40$, respectively) and the male silver eel stage (MII, $n = 7$).

## Data analysis
### Head-width distribution
ImageJ (*Abràmoff, Magalhães & Ram, 2004*) was used to measure head width (HW) on the photographs as two times the snout length, which is defined as the distance from the midpoint between the anterior end of the eyes to the tip of the snout (Fig. 2). This way, HW was measured at the postorbital region where the jaw muscles can be found, an important region related to broad- and narrow-headedness (*De Meyer, Christiaens & Adriaens, 2016*). In addition, head length (HL) was measured as the distance from the tip of the snout to the start of the pectoral fins and consequently, HW/HL was calculated for each eel. Since HW/HL tends to increase slightly with TL, the unstandardized residuals were first calculated via linear regression between HW/HL and TL  (see Appendix for more details).

Verhelst et al. (2018), *PeerJ*, DOI 10.7717/peerj.5773

**Table 1** **Numbers of eels caught per maturation stage with the different morphometrics: total length (TL), body weight (BW), left horizontal and vertical eye diameters (EDh and EDv, respectively) and left pectoral fin length (FL).** Means ± SD (range) are given.

| Stage | Number | TL (mm) | BW (g) | $ED_h$ (mm) | $ED_v$ (mm) | FL (mm) |
|---|---|---|---|---|---|---|
| I | 51 | 345 ± 76 (184–501) | 76 ± 46 (9–222) | 4.11 ± 0.97 (2.01–5.76) | 3.84 ± 0.92 (1.67–5.39) | 15.42 ± 3.78 (7.88–25.44) |
| FII | 68 | 499 ± 47 (426–642) | 213 ± 76 (88–478) | 5.93 ± 0.48 (4.66–7.02) | 5.51 ± 0.46 (4.59–6.65) | 23.22 ± 2.50 (16.68–29.98) |
| FIII | 91 | 639 ± 78 (505–835) | 504 ± 199 (141–1106) | 7.65 ± 0.70 (6.28–9.08) | 7.14 ± 0.69 (5.46–9.70) | 30.38 ± 3.78 (24.24–40.32) |
| FIV | 15 | 815 ± 67 (707–932) | 1173 ± 248 (771–1830) | 10.43 ± 0.81 (9.31–12.49) | 9.76 ± 0.79 (8.91–11.86) | 41.17 ± 4.54 (30.84–48.18) |
| FV | 40 | 630 ± 70 (510–775) | 502 ± 177 (189–912) | 8.86 ± 0.94 (7.40–11.18) | 8.40 ± 0.90 (6.95–10.39) | 32.80 ± 4.03 (25.84–45.37) |
| MII | 7 | 386 ± 3 (335–428) | 111 ± 39 (66–170) | 6.69 ± 1.26 (4.47–8.16) | 6.22 ± 1.09 (4.27–7.52) | 20.06 ± 3.89 (12.97–25.75) |

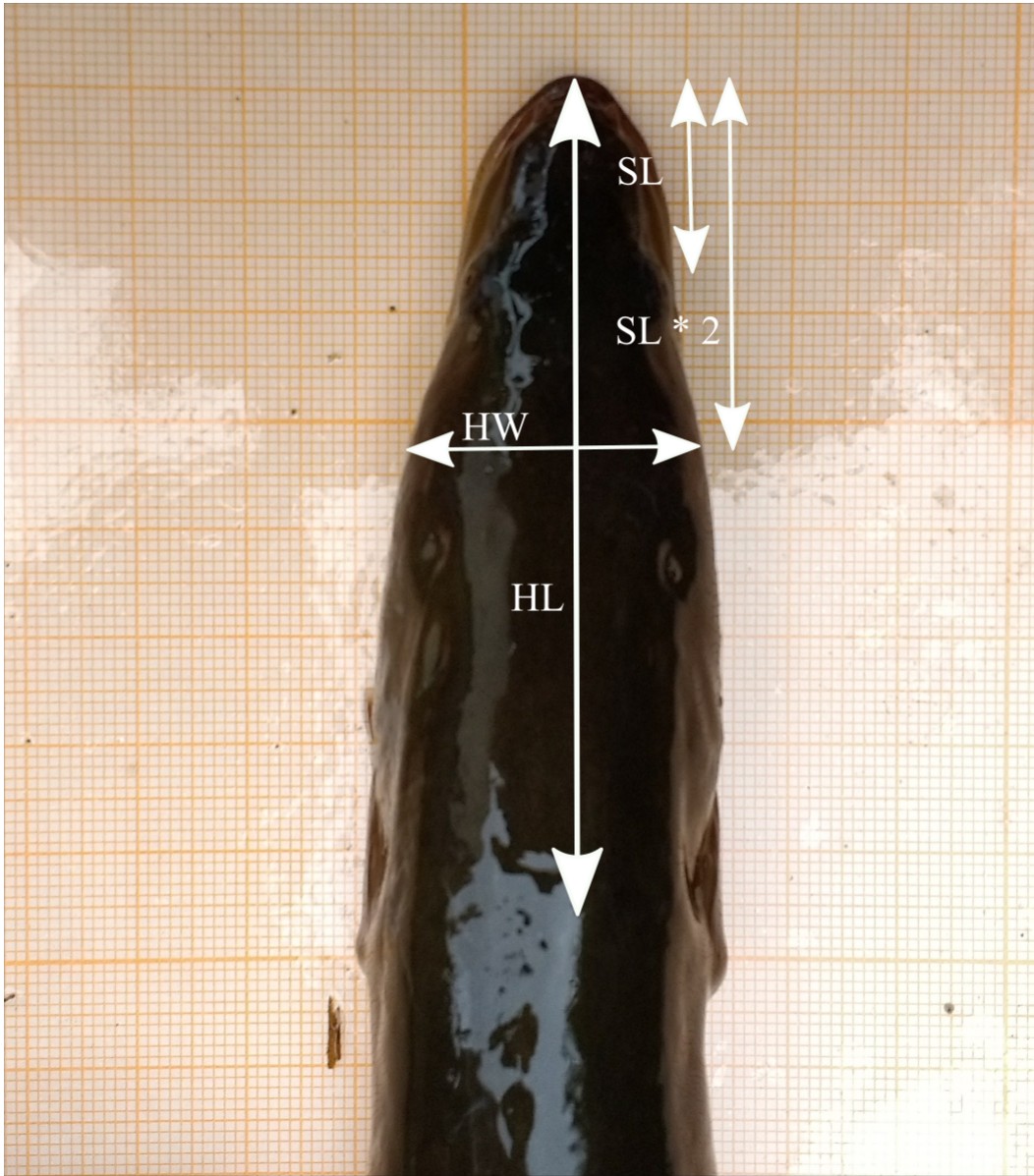

**Figure 2** Head measurements based on the dorsal picture of an eel's head on graph paper (HL, head length; HW, head width; SL, snout length) (photo credit: Pieterjan Verhelst).

Subsequently, the residual values were used for a mixture analysis in the R environment (*R Development Core Team, 2017*). To analyze whether the head shape variation followed a unimodal or bimodal distribution, two different penalized model selection criteria were calculated: the Akaike Information Criterion (AIC) and the Bayesian Information Criterion (BIC). Both model selection criteria are commonly applied with lower values indicating better models, but have different qualities and merits (*Aho, Derryberry & Peterson, 2014*). Essentially, AIC is applied when the analysis is exploratory and strives for efficiency, that is, the method maximizes predictive accuracy. Consequently, AIC tends to select the most

complex model as the true model (*Kass & Raftery, 1995*). BIC on the other hand is used for confirmatory analysis and strives for consistency (*Aho, Derryberry & Peterson, 2014*). Related to unimodal and bimodal distribution selection, according to *Brewer (2003)*, a unimodal distribution is strongly and moderately supported when $\Delta$AIC $< -8$ and $< -5$, respectively. If $\Delta$AIC ranges from $-5$–$5$, there is equal support for both a unimodal and bimodal distribution, while values $>5$ and $>8$ moderately and strongly support bimodality, respectively. We used the 'mclust' package of the R environment for model selection criterion calculation, and the 'mixtools' package for visualizations (*R Development Core Team, 2017*).

### Maturation stages and sex

First, we checked if the unimodal distribution held true for the different maturation stages (I, FII, FIII, FIV, FV and MII) separately by conducting a one-way ANOVA on the residual variance of each maturation stage. Next, the AIC and BIC were calculated for each maturation stage as mentioned above.

### Body condition

To analyze if body condition changes according to HW, the relative condition factor ($Kn$) (*Le Cren, 1951*) was used. $Kn$ takes allometric growth into account; when $<1$, fish are in a worse condition than expected, while $>1$ indicates a better condition:

$$Kn = \frac{W}{aL^b}$$

where a is a constant and b an exponent varying from 2.5 to 4 (*Hile, 1936*; *Martin, 1949*): $b = 3$ indicates isometric growth and b $\neq$ 3 allometric growth (b $<$ 3 for fish becoming more fusiform as they grow and b $>$ 3 for fish becoming progressively less slender). In the formula, total length (L) and body weight (W) have a logarithmic relationship:

$$LogW = loga + b * logL$$

where $b$ is the slope of the line and log a the intercept (*Le Cren, 1951*). To test if Kn changes according to HW, linear regression was applied (data followed a normal distribution and the variances were homogenous).

### Migration speed

To determine migration speeds, 51 migrating eels were tagged with coded acoustic transmitters (V13, 13 $\times$ 36 mm, weight in air 11 g, frequency 69 kHz, estimated battery life: 1,021–1,219 days (battery lifetime depended on specific transmitter settings)) from VEMCO, Ltd. (Canada, http://www.vemco.com) and tracked in the Zeeschelde by an acoustic network of 25 acoustic listening stations (ALSs) (VR2W; VEMCO Ltd., Beford, Canada) (approval by the Ethical Committee of the Research Institute for Nature and Forest (ECINBO09)). After anaesthetizing the eels with 0.3 ml L$^{-1}$ clove oil, tags were implanted according to *Thorstad et al. (2013)* with permanent monofilament. Eels recovered in a quarantine reservoir for approximately one hour and were subsequently released at the ALS closest to their catch location. Data were processed as previously described in *Verhelst et al. (2018a)*. The residency times (i.e., the time between arrival and departure at an ALS)

were calculated, which allowed us to reduce the data by accumulating the number of detections during a fixed period of time. We applied an absence threshold of one hour (i.e., the maximum time permitted between detections within a single residency period) and a detection threshold of one detection (i.e., the minimum number of detections required for a residency period). As such, the residency search resulted in intervals with arrival and departure times per eel at each ALS.

Not all eels migrated upon tagging. Therefore, an eel was considered migratory when it travelled net $\geq$ 20 km downstream during $\leq$ 40 days (*Verhelst et al., 2018b*). Within that period, we selected the records from the most upstream station down to the most downstream station (i.e., sometimes an eel aborted its migration and moved back upstream). The 20-km threshold is based on the maximum range distance found for yellow eels (i.e., 18 km) (*Verhelst et al., 2018c*) plus two times the one km detection range of an ALS in the Zeeschelde (i.e., the spatial error for the migration range). The 40-days threshold is based on the finding that eels not migrating net $\geq$ 20 km downstream during that period, arrested their migration to proceed in a next season. For two eels, applying the above assumptions resulted in the selection of two migration phases per eel: they arrested their migration, subsequently moved back upstream near their catch location, and eventually resumed migration two and twelve months later. For those two eels, we only used the second migration phase for analysis. Next, we calculated the migration speed as the time needed to cross the distance between the detections at the two most distant ALSs in the migration phase. To analyze if the migration speed differed according to HW, a linear mixed effects model (transmitter ID as a random effect to account for autocorrelation) was applied. We also applied the linear mixed effects model after removal of three extreme values. The nlme R package was used to conduct the linear mixed effects model (*R Development Core Team, 2017*).

## RESULTS

### Head-width distribution
The linear regression of the HW/HL ratio to TL proved significant ($F_{(1, 270)} = 51.26$, $p = 7.66e^{-12}$ with $R^2$ (adjusted) = 0.16), and revealed the following relationship (Fig. 3):

$$HW/HL \sim 0.26244 + 0.00087 * TL.$$

The data followed a normal distribution (Shapiro–Wilk test, $W = 0.99$, $p > 0.05$), yet showed slightly right-tailed skewness. BIC proved lowest for the unimodal distribution, favoring that distribution. AIC on the other hand was lowest under the bimodal distribution, but differences between unimodality and bimodality were consistently small (Table 2). Moreover, when using the criteria of *Brewer (2003)*, our data provided equal support for both unimodality and bimodality under AIC, since $\Delta$AIC ranged between $-5$ and $+5$. However, visualization of the bimodal distribution indicated a strong overlap between the two normal distributions (i.e., one normal distribution is almost completely encompassed by the other) (Fig. 4). Based on these results, we concluded that a unimodal distribution best fitted our data.
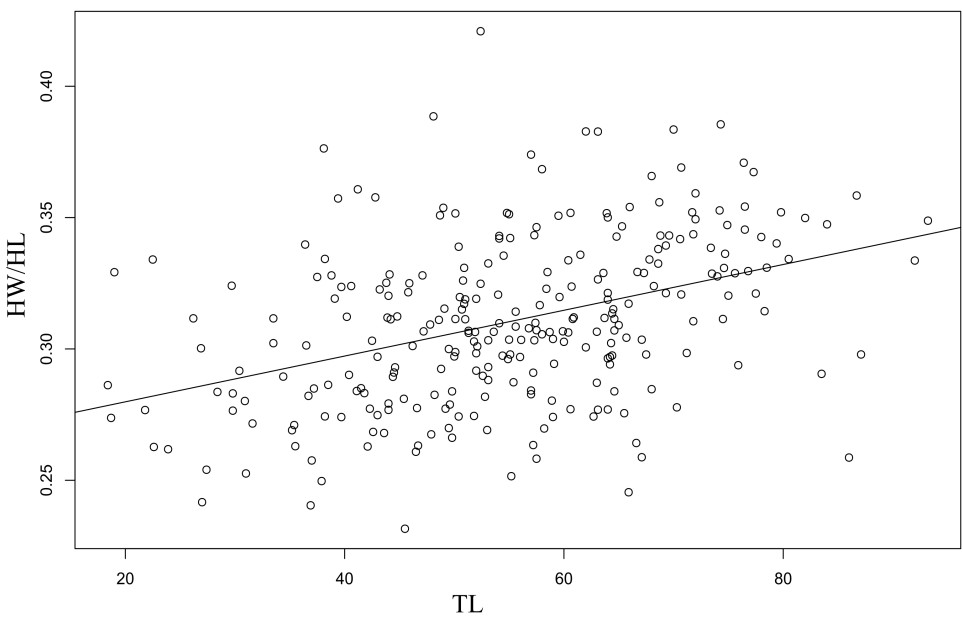

**Figure 3** Regression between the ratio head width : head length (HW/HL) and total body length (TL).

**Table 2** Values of the model selection criteria AIC and BIC for a unimodal and bimodal distribution.

| Model selection criterion | Unimodal | Bimodal |
| --- | --- | --- |
| AIC | −1,148 | −1,149 |
| BIC | −1,141 | −1,134 |

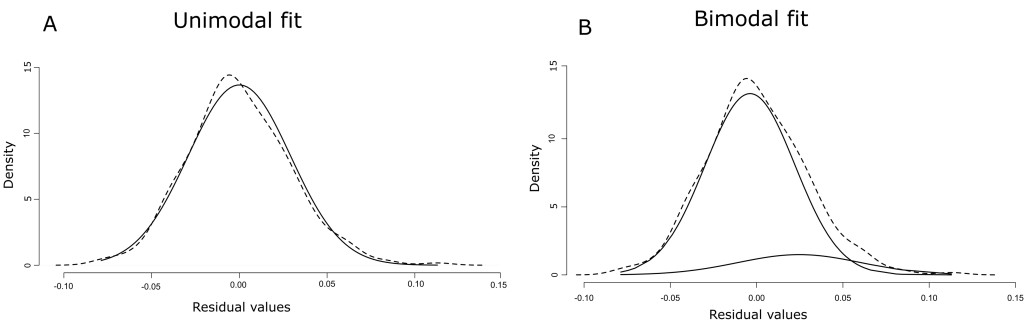

**Figure 4** Unimodal (A) and bimodal fit (B) of normal distributions (solid lines) on the density distribution of the residuals (dashed lines).

## Maturation stages and sex

We did not find a significant difference in residual variation between the different maturation stages (one-way ANOVA, $F = 0.83$, DF = 5, $p > 0.05$), although the variation for MII eels, which was based on only seven individuals, was slightly higher than for the
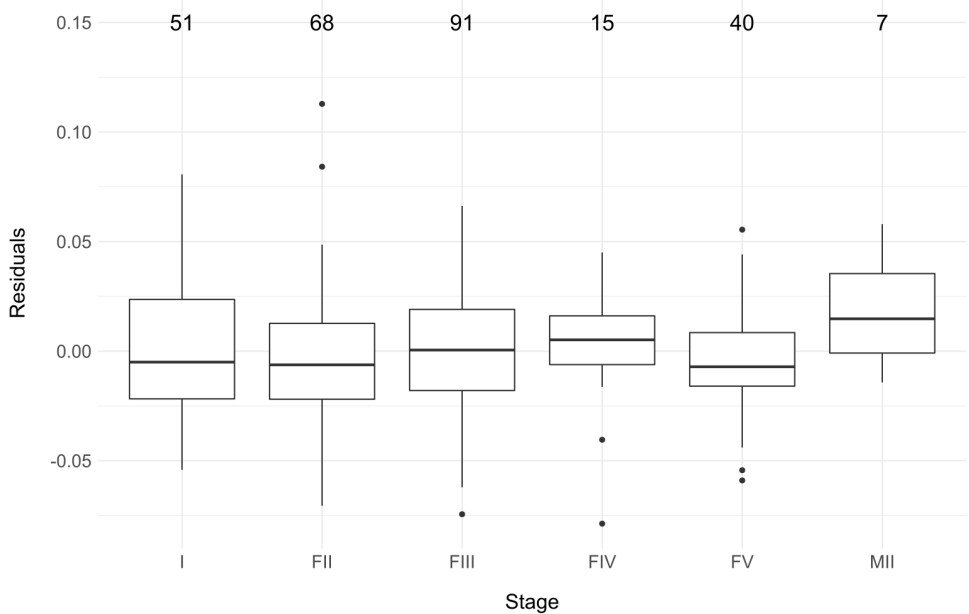

**Figure 5  The residual variation according to the six maturation stages (I, FII, FIII, FIV, FV and MII).** The number of eels per stage are indicated above the boxplot.

**Table 3  The AIC and BIC per maturation stage (I, FII, FIII, FIV, FV and MII) for both unimodal and bimodal support.**

| Stage | Unimodal | | Bimodal | |
|---|---|---|---|---|
| | AIC | BIC | AIC | BIC |
| I | −206 | −202 | −208 | −200 |
| FII | −282 | −277 | −288 | −279 |
| FIII | −384 | −379 | −380 | −370 |
| FIV | −59 | −58 | −60 | −57 |
| FV | −175 | −171 | −171 | −164 |
| MII | −28 | −28 | −30 | −30 |

other groups (Fig. 5). Similar to the total dataset and following the guidelines of *Brewer (2003)*, BIC favored the unimodal distribution for all stages except FII and MII, while uni- and bimodality were equally supported by AIC between eel stages (Table 3). Yet again, there was a strong overlap between the two normal distributions under the bimodal model (Fig. 6). Notably, due to the low number of observations, especially for FIV- and MII-eels, more data is needed to draw strong conclusions on the life stages.

## Body condition

Values for the constants $a$ and $b$ of the logarithmic relationship between weight and total length were $a = 0.00068$ and $b = 3.24$,

$$Kn = \frac{W}{0.00068 \, L^{3.24}}$$

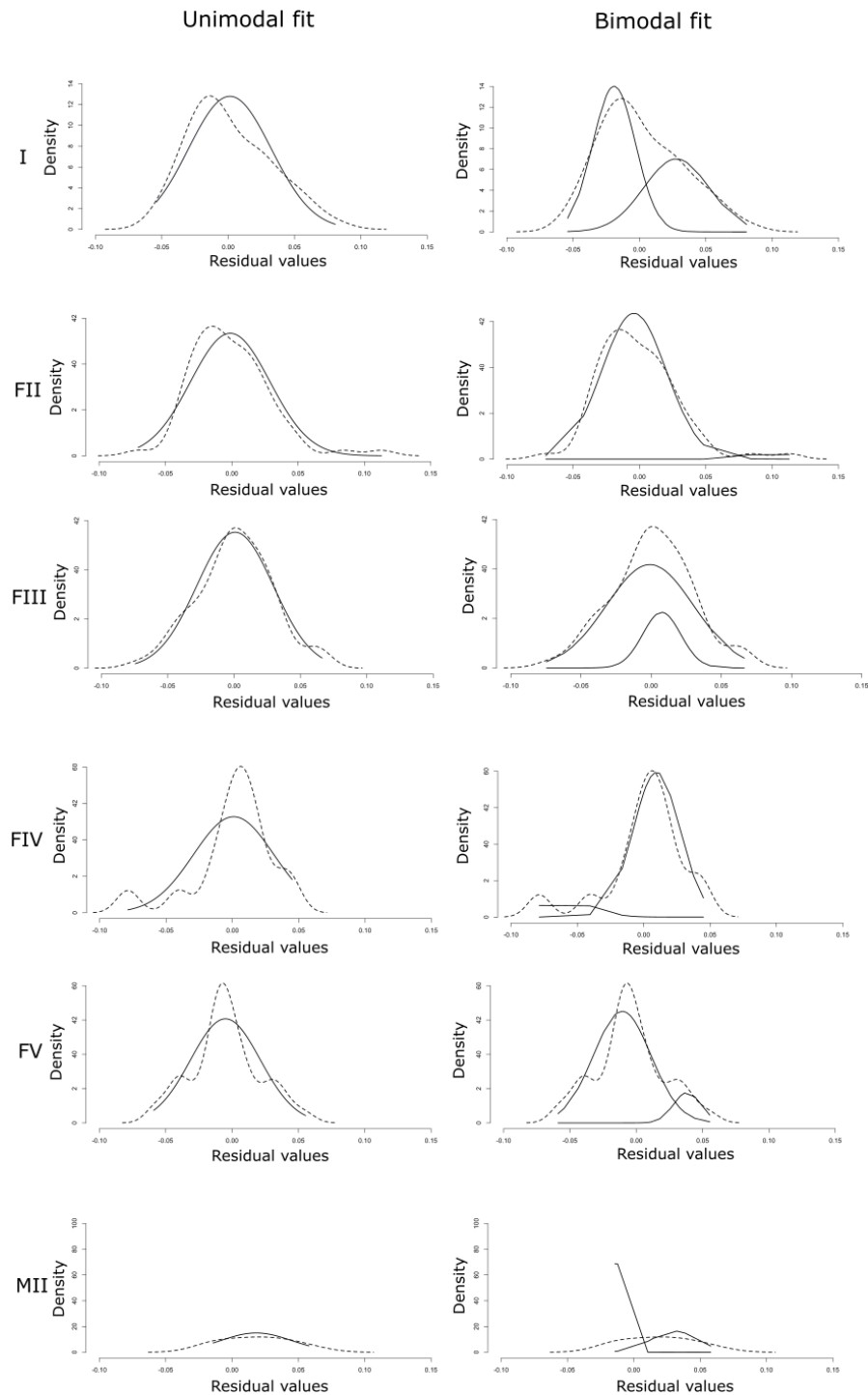

**Figure 6** Unimodal and bimodal fit of normal distributions (solid lines) on the density distribution of the residuals (dashed lines) for each maturation stage (I, FII, FIII, FIV, FV and MII).

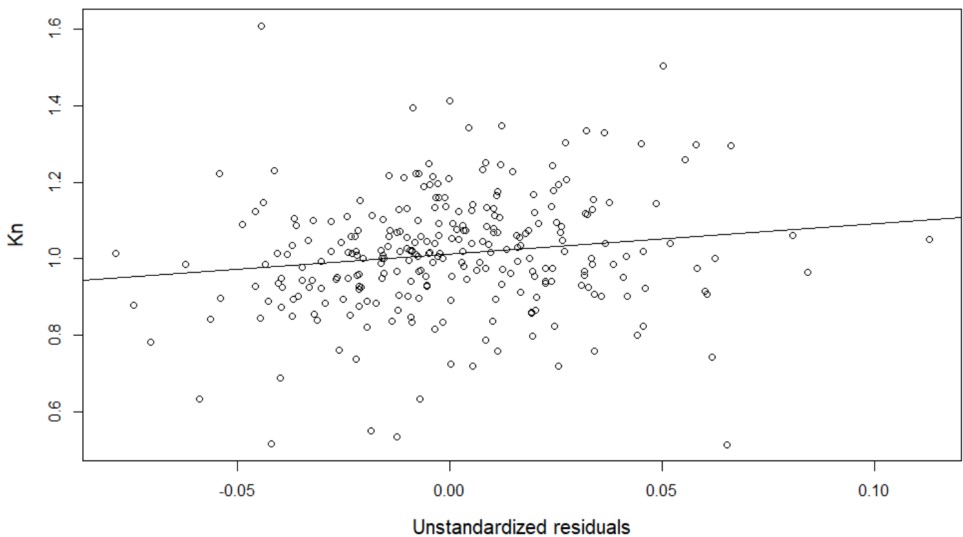

**Figure 7** **The relative condition (Kn) increases with a broader head width (unstandardized residuals).**

indicating that eels become plumpier as they grow (b > 3). Kn was on average $1.01 \pm 0.15$ (range: 0.51–1.61) and increased significantly with a broader HW (linear regression, $F(1, 270) = 6.30$, $p = 0.01$ with $R^2$ (adjusted) = 0.02) (Fig. 7):

$$Kn \sim 1.01 + 0.80 * \text{unstandardized residuals}.$$

### Migration speed

Migration speed was on average $0.05 \pm 0.08$ m s$^{-1}$ (range: 0.01–0.40 m s$^{-1}$) and did not change significantly according to HW (linear mixed effects model, $t$-value 0.63, $DF = 49$, $p = 0.53$; Fig. 8), not even after removal of the three outliers (linear mixed effects model, $t$-value 1.14, DF = 46, $p = 0.26$).

## DISCUSSION

### Head-width distribution

Despite the dichotomous characterization of eel HW in previous research based on eels from multiple locations and/or habitats (*Ide et al., 2011*; *Proman & Reynolds, 2000*), our study at a single location in the Zeeschelde does not support clear bimodality and hence also does not provide any indication for disruptive selection. Instead, BIC indicated unimodality and AIC provided equal support for a unimodal and a bimodal distribution (*Brewer, 2003*). Nonetheless, AIC tends to select the more complex model over the true model (*Kass & Raftery, 1995*). Indeed, the equal support for both unimodality and bimodality is likely caused by the strong overlap between the two normal distributions in the bimodal model, with one normal distribution being almost completely encompassed by the other. Such overlap can hamper the distinction between a unimodal and a bimodal distribution (*Hendry et al., 2006*). Due to this strong overlap, we conclude that eels in the present study cannot be strictly classified into narrow- and broad-headed individuals based on a single

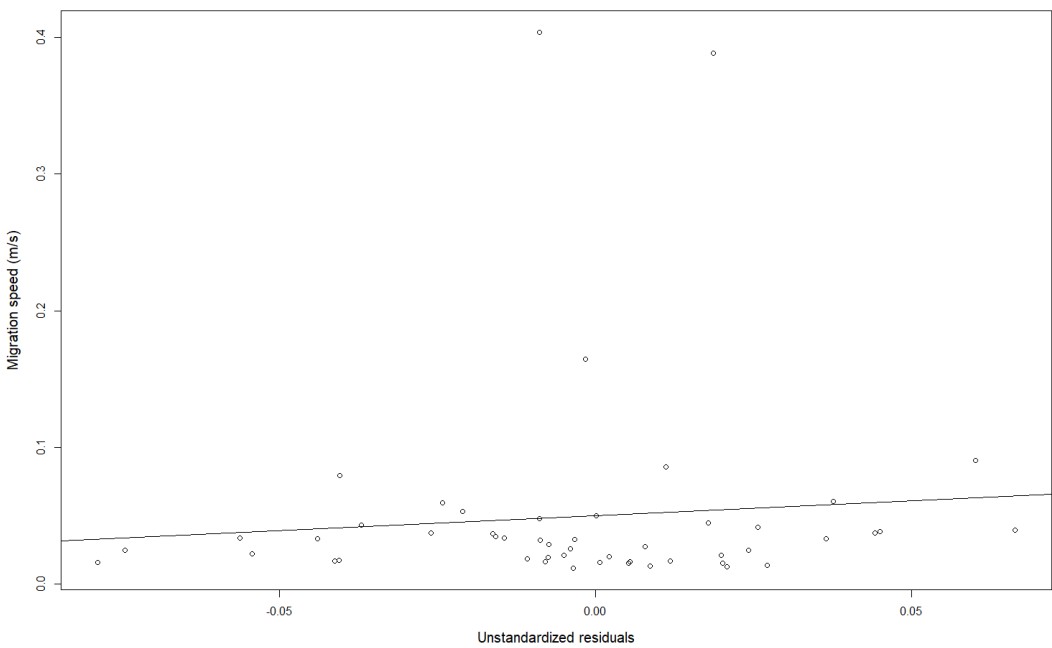

**Figure 8 Migration speeds in relation to the head width (unstandardized residuals).**

threshold (*Barry et al., 2016*; *Ide et al., 2011*; *Lammens & Visser, 1989*; *Proman & Reynolds, 2000*). Instead, a unimodal distribution indicates that eels have narrower or broader heads towards the extremes of a continuous normal distribution with many intermediate morphs. Notably, the slight right-skewness in the HW variation in the present study may be attributed to other selective pressures than disruptive selection. For instance, these data could be interpreted as an indication for a unidirectional pressure towards larger head widths, perhaps reflecting selection for predation on larger or hard-bodied prey. As such, skewness in one direction or the other may vary widely between locations and habitats. Although the number of eels in our study was relatively limited ($n = 272$), analysis of 50%, 75% and 90% of the data yielded very similar results (Fig. S1, Table S1). Moreover, the overlap between the two normal distributions under the bimodal fit tended to increase with the percentage of data taken into account (i.e., 50% to 90%), indicating a stronger support for unimodality as more data was taken into account. In addition, other studies have used similar or even lower numbers (*Barry et al., 2016*; *Cucherousset et al., 2011*; *Kaifu et al., 2013*; *Proman & Reynolds, 2000*).

*Ide et al. (2011)* did find evidence of bimodality and observed that head shape variation in European eel in Belgium was best described by two unimodal distributions with overlapping tails. This discrepancy may be explained by the fact that these authors covered different sampling locations, often characterized by different feeding conditions. If head shape depends on prey type, then eels caught at locations with a higher abundance of soft-bodied/small prey will tend to the narrow side of the HW distribution, while the opposite will hold true for locations dominated by hard-bodied/large prey. When eels of

two such contrasting locations are pooled together, a bimodal distribution would be more likely to occur.

Under the assumption that HW distribution is mainly the result of food choice (*Lammens & Visser, 1989*; *Proman & Reynolds, 2000*), the observed unimodal distribution in the Zeeschelde could be explained by an opportunistic behavior of eels (*Lammens et al., 1985*; *Schulze et al., 2004*; *Van Liefferinge et al., 2012*). Feeding on a wide range of prey items reduces selective pressures towards head shapes that are more specialized for the consumption of either hard or soft prey. Predatory fish of cold-temperate waters tend to be opportunistic feeders, as productivity in these areas is often relatively low and prey abundance depends on season and temperature (*Keast, 1979*), implying that the most available prey has the highest chance of being consumed. However, eels can also display a remarkable preference for specific prey items, irrespective of their availability (*Barak & Mason, 1992*).

Other factors than food could also explain the occurrence of head dimorphism: narrow headed eels have been suggested to be more crepuscular and forage in the littoral zone, while broad headed eels would be more active at night and in the limnetic zone (*Barry et al., 2016*; *Cucherousset et al., 2011*). In addition, bimodality may be present mostly in areas where eel densities are high, leading to intraspecific competition through resource polymorphism and consequently to different head shapes (e.g., in lakes with artificially stocked eels) (*Lammens & Visser, 1989*).

## Maturation stages, sex and body condition

Eel maturation stages are commonly classified according to *Durif, Dufour & Elie (2005)*; *Barry et al. (2016)*; *Bultel et al. (2014)*; *Stein et al. (2015)*. Although the method may not be 100% conclusive, distinction between male and female silver eels was confirmed in our study as males showed the typical silvering characteristics (visible lateral line, large, melanised pectoral fins, dark dorsal side, silver-white ventral side and large eyes) and had a TL <45 cm (*Tesch, 2003*).

*De Meyer et al. (2015)* hypothesized that the absence of a clear bimodal pattern in glass eels, contrasting with its presence in yellow eels (*Ide et al., 2011*), may be attributed to a trophic niche segregation between different eel developmental stages. However, we found no bimodal pattern in the Zeeschelde in any of the maturation stages defined by *Durif, Dufour & Elie (2005)*. Like for the total dataset, BIC favored a unimodal distribution and AIC provided equal support for a unimodal and a bimodal distribution. Again, the latter likely results from the strong overlap between two normal distributions. Given the small number of specimens in the present study, especially in FIV (15) and MII (7) eels, we can, however, not rule out the possibility that the distribution could be skewed due to the tail of the distribution (*Hendry et al., 2006*). The absence of a clear bimodal distribution could again be explained by the opportunistic behavior of the eels (*Lammens & Visser, 1989*; *Schulze et al., 2004*; *Van Liefferinge et al., 2012*). Specifically, since our study included eels from a single location only, opportunistic feeding and low to moderate population density would render disruptive selection pressure towards feeding specificity unlikely during the different maturation stages in the Zeeschelde.

Counter to *Cucherousset et al. (2011)*, who argued that the better body condition of both narrow and broad headed eels compared to intermediate headed eels was the result of disruptive selection (*Martin & Pfennig, 2009*; *Skulason & Smith, 1995*), body condition of eels in the Zeeschelde also did not support the idea of disruptive selection, since body condition increased along with HW, suggesting unidirectional selection. However, the small amount of variation explained by the model suggests that factors other than head width play a more prominent role in body condition variation.

### Migration speed

Combining telemetry with HW classification, *Barry et al. (2016)* observed a larger home range for broad headed yellow eels. In addition, circadian activity patterns differed, with narrow-headed yellow eels being more crepuscular while broad-headed yellow eels more nocturnal. Here, we preliminarily analyzed if the downstream migration speed (i.e., movement at meso-scale) of silver eels in the Zeeschelde differed according to HW. Migration speed is often calculated to make predictions about progression (*Aarestrup et al., 2010*; *Breukelaar et al., 2009*; *Bultel et al., 2014*), swimming performance (*Russon, Kemp & Calles, 2010*; *Van Den Thillart et al., 2004*; *Van Ginneken et al., 2005*) or the chances of reaching the spawning area in time (*Righton et al., 2016*). Our results suggest that at least the progression of silver eels is not influenced by their head morphology. Nonetheless, swimming experiments in swim tunnels may shed more light on the relationship between HW and different aspects of migration and swimming performance (*Van Ginneken et al., 2005*).

## CONCLUSION

In contrast to evidence for a bimodal head-width distribution of European eel (*Ide et al., 2011*), we found support for a unimodal distribution in European eel HW variation at a location in the Zeeschelde, both when separately analyzing different maturation stages and when looking at the total dataset. This indicates a lack of evidence for disruptive selection but does not exclude unidirectional pressures on variation in eel head shapes. Finally, downstream migration speed of silver eel at a meso-scale was not influenced by HW morphology. We conclude that eels in the Zeeschelde could not be dichotomously classified into narrow and broad heads, but rather represent a continuum of specimens with narrow to broad heads following a normal distribution.

## ACKNOWLEDGEMENTS

We would like to thank R Baeyens, S Bruneel, N De Maerteleire, S Franquet, E Gelaude, T Lanssens, S Pieters, K Robberechts, T Saerens, R van der Speld and Y Verzelen who assisted with the data collection. We also want to thank D Buysse and J Van Wichelen whose comments helped to improve the manuscript.

### Funding

Pieterjan Verhelst holds a doctoral grant from the Flemish Agency for Innovation & Entrepreneurship (VLAIO), now under the auspices of the National Science Fund FWO. This work was supported by data and infrastructure provided by the INBO and VLIZ (RV Simon Stevin and RHIB Zeekat) as part of the Flemish contribution of the LifeWatch ESFRI observatory. The funders had no role in study design, data collection and analysis, decision to publish, or preparation of the manuscript.

### Grant Disclosures

The following grant information was disclosed by the authors:
Flemish Agency for Innovation & Entrepreneurship (VLAIO).
National Science Fund FWO.
INBO.
VLIZ.

### Competing Interests

The authors declare there are no competing interests.

### Author Contributions

- Pieterjan Verhelst conceived and designed the experiments, performed the experiments, analyzed the data, contributed reagents/materials/analysis tools, prepared figures and/or tables, authored or reviewed drafts of the paper.
- Jens De Meyer conceived and designed the experiments, analyzed the data, authored or reviewed drafts of the paper.
- Jan Reubens performed the experiments, contributed reagents/materials/analysis tools, authored or reviewed drafts of the paper.
- Johan Coeck contributed reagents/materials/analysis tools, authored or reviewed drafts of the paper.
- Peter Goethals authored or reviewed drafts of the paper.
- Tom Moens and Ans Mouton authored or reviewed drafts of the paper, approved the final draft.

### Animal Ethics

The following information was supplied relating to ethical approvals (i.e., approving body and any reference numbers):

The Ethics Committee of the Research Institute for Nature and Forest has reviewed and discussed the application to conduct the animal experiment (ECINBO09).

### Data Availability

GitHub: https://github.com/PieterjanVerhelst/headdimorphism.

## Supplemental Information

Supplemental information for this article can be found online at http://dx.doi.org/10.7717/peerj.5773#supplemental-information.

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
