# Peer review of "Unimodal head-width distribution of the European eel (Anguilla anguilla L.) from the Zeeschelde does not support disruptive selection"

_PeerJ, doi:10.7717/peerj.5773_

## Round 0.1 · original submission · Major Revisions

Your manuscript has now been assessed by two expert reviewers. We all agree that the question addressed in the article is interesting, but have all also expressed some concerns about the analysis and interpretation of data. There are three main issues:

1) I note several potential issued with statistical analysis in the current manuscript, namely the rather unconventional approach of using 3 different model selection 'rules'. You do not not justify why you take this approach, nor what you would do if they disagree. In fact, they do disagree and you appear to simply pick one of the potential outcomes as the de facto result. Given these results, I am surprised that you have chosen to title your paper as you have.

2) I also question whether you have sufficient data to test for bimodality among so many different subdivisions. Density histograms of raw data might convincingly show unimodality, but might also be completely the wrong shape relative to the unknown 'truth' of all eels because they are based on so few data when you have divided 272 eels into 6 categories.

3) From a broader perspective, I worry about your attempts to generalize results from this study and single location to all European eels, as reflected by your manuscript title. Perhaps the one location you have chosen to study shows no evidence of bimodality (though AIC/BIC say otherwise), but that doesn't mean it's a universal truth for all European eels. You will not the same concerns raised by the reviewers.

Comments from myself and the two reviewers are appended below. I invite you to revise your article in line with these comments, but note that if I find the revisions to be unsatisfactory I will unfortunately have no choice to reject the resubmission without further peer review.

Editor's Comments:
Line 75: “genetic evidence for this hypothesis” is not a velar statement. I think you mean “supporting this hypothesis”, but unfortunately the phrase “genetic evidence” is in itself very vague. Please rephrase.

Line 141/142: These may not be residuals from mass on length, but residuals of the head width/head length ratio over tail length will still suffer from the same problems, namely that these kinds of regressions are problematic when there is measurement error on the X axis. See Peig & Green 2009 Oikos 12, 1883-1891 for a discussion

Line 145-146: Why use three different model selection criteria? You don’t explain the rationale here, nor what you’d do if they disagreed – which is likely since log likelihood doesn’t penalise models for complexity, and BIC/AIC do so but to different extents depending on sample size. In fact, that’s exactly what happens (Results line 210), with more formal model selection criteria in fact favouring bimodality.

Line 152-156: I think the question here, whether unimodality varies by maturation stage, is both valid and interesting. But I do not believe you have anywhere near the sample size to answer this question from your data. 272 eels might give a reasonable approximation of the true population distribution if there were no subdivisions, but you are testing 6 such groupings here, some with single-digit sample sizes.

Migration speed: I’m not convinced that acoustic tags are the best tool to allow you to answer some of your questions. The same weight of tag is likely to represent a different relative encumbrance depending on the weight/size/shape of the eel. At best you might be able to show that different head sizes of eel have different life history strategies, but it would allow you to test whether certain eel head shapes are more hydrodynamic (which I think is what you’re getting at with hypothesis 3). That is, narrow-headed eels might be capable of faster swim speeds because of hydrodynamics, but wild tracking might not reveal
Put another way, narrow-headed eels might incur lower energy costs for swimming the same distance at the same speed, but might not actually swim any faster. I think you’d need to do some experimental work with a flow chamber to test these hypotheses.

Line 187: “Not all eels migrated and those that did, not always migrated upon tagging”. It’s not very clear what you mean here. Presumably just that not all tagged eels migrated, in which case this sentence seems to be saying the same thing twice ( I may be wrong). More importantly, if not all tagged eels did migrate then this analysis doesn’t state the final sample size used, as presumably these eels would not be included.

Lie 215-216: Strong overlap between normal distributions does not mean bimodality is not present or supported (even though AIC does support bimodality).

Figure 5: maturation stages not defined in the legend.

Figure 8: there are no units for the speed on the y axis of the graph.

Reviewer 1 ·

Basic reporting

• The article is generally clear and well written and has only one unexpected word choice; “plumpier” may perhaps better be written as “more plump”. That may well be a matter of choice as some others would prefer “plumper”. Here, however, I would suggest “less slender” as eels with their “anguilliform shape” [here in the locomotory sense as coined by Breder 1926] aren’t “plumpy“.

• The scientific background literature of the Introduction is reasonably complete. I could only find one important omission of relevance to the MS:

Muschick et al. 2011. Adaptive phenotypic plasticity in the Midas cichlid fish pharyngeal jaw and its relevance in adaptive radiation. BMC Evolutionary Biology, (2011) 11:116. (https://doi.org/10.1186/1471-2148-11-116)
(https://bmcevolbiol.biomedcentral.com/articles/10.1186/1471-2148-11-116).

The article by Muschick et al. (2011) is far more relevant than the cited general articles on bats (Saunders & Barclay 1992) or the ecomorphology of crocodylians (Iijima et al 2017). The other articles cited refer to interpretations of ecomorphology (Hahn & Cunha 2005, Narayani et al 2015, Norton & Brainerd 1993) are both relevant and interesting, but remain interpretative and not really conclusive.

The Muschick et al. (2011) article demonstrate using a controlled experimental design how different food items change head morphology. Much of the current manuscript’s text appears to assume the reverse, that different head morphologies lead to different feed strategies. I am not convinced that is the case here. A shift in perspective would permeate how the wordings of the entire manuscript are chosen and written as well as how previous research may and can be interpreted.

This is a very important difference in causality and must be addressed far more clearly by the authors in the current manuscript.

• The structure of the article conforms to an acceptable format as per the journal’s standards.

• Figures are relevant to the article, even if Figure 6 has diminutive fonts and is hard to read.

• There are no raw data, but the abundance of graphs should make them superfluous.

• The manuscripts is self-contained with results relevant to the four (sub-)hypotheses tested.

• There are four research questions in this manuscript which are reciprocally relevant to each other.


.

Experimental design

• It is my impression that this is original primary research within the Aims and Scope of the journal.

• The four stated research questions are well defined, relevant and meaningful on their own. Later, the paper states that the results challenge the results and conclusions of several other recent papers. However, the Introduction may perhaps be better divided into paragraphs better reflecting the backgrounds of the four distinct research questions; the Discussion already is divided as per those research questions. Still, as explained below, they may also be seen as four sub-hypotheses of another more encompassing question about “disruptive selection”. That question would also be of more general interest.

• One should note that the ecological and evolutionary interpretations suggested in other articles and duly referred to in the Introduction also are discussed by the authors. Still, they are apparently not considered “research question” even if they make up much of the discussion. Here I think of

◦ resource polymorphism
◦ resource selectivity
◦ phenotypic plasticity
◦ disruptive selection

In fact, it may be the most important conclusion of this paper that they have no evidence for disruptive selection as has been suggested in other papers. One of the main arguments here is then that they have examined only one river system and that there is no bimodality. Other articles have compared allopatric eels, from two different river systems where a pooled bimodality then rather reflects the differences between rivers (biotic/abiotic) than any “disruptive selection”. That, I find, is an important aspect of this manuscript and should perhaps even be reflected in the title:

“Unimodal head width distribution of the European eel (Anguilla anguilla L.) from a single river drainage does not support disruptive selection”.

• It is my impression that the reliability of the presented results is high. The number of observations is 272 which the authors themselves suggested (at line 250) might be on the low side. Yet, given the simplicity of the research questions it is my conviction that the reliability remains high. Had this been a multivariate analysis it would likely have been too few observations, and, an initial power analysis should then have been performed as well. Here that shouldn’t have been needed.

• The research has been conducted in conformity with the prevailing ethical standards in the field, explicitly referring to that at lines 175-176.

• The material and methods used have been described with sufficient details to be reproducible by other investigators.


.

Validity of the findings

• The authors of the current manuscript challenges previous research, by finding that bimodality is not a general phenomenon to eels. That is a key finding.

The results may at first be seen as trivial as the authors have only investigated one river system leading to unimodality and dismissed other authors’ results because being from two different river systems. The authors interpret this lack of bimodal as a better way to analyze “disruptive selection” in eels. I fully agree with that approach. I think that “disruptive selection” can be seen as a full research question on its own, with the current four research questions as “sub-hypotheses”.

The authors write (at lines 273-274): ”The equal support for both unimodality and bimodality is likely caused by the strong overlap between the two normal distributions in the bimodal model”. That wording suggests the authors believe(d) statements like “this narrow-headed eel is really a broad-headed, contrary to factual observation”. It suggests the authors may have had an initial difficulty in accepting their own observations. I find that strange and perhaps it should be reworded as it may appear sarcastic. Still, the conclusions that follow suggest that narrow-headed eels are narrow-headed eels. That is good.

• I have not seen the raw data and have no reason to believe they are wrong. I don’t think they are necessary here. The statistical methods used are well chosen and good enough for the purpose. The number of observations most likely are too few for several forms of multivariate analyses.

• My overall suggestions are that the authors make a clearer distinction between the perspectives:

- Head morphology leading to food items
- Food items leading to head morphology

Here, the term “food items” can be replaced with “swimming performance”, “habitat choice” etc. The causality perhaps remains obscure either way as the manuscript did not include stomach analysis, but … you can’t have it all.

• The article does challenge the conclusions of other articles and I think the authors have demonstrated that many statements in literature lack substance. Yet I think the wordings and the Discussion needs to be rewritten and expanded with more perspectives regarding correlation and causality.


.

Reviewer 2 ·

Basic reporting

This paper is well written and the language and grammar are good. I have marked some edits on the manuscript.
The paper addresses one of the longstanding topics of eel biology, their headshape. The debate has long focussed on shape and diet and competitive advantages, adaptation vs survival.
In my view, this paper doesn’t make much more progress in untangling the cause effect of diet and headshape. The additional information on maturation, condition and migration are interesting, but it is admitted in the paper that the differences are small or absent and sample sizes are in places too small. However, the title is Unimodal head width distribution….. even though the tests on lines 205-216 are largely inconclusive, one pointing to unimodality and two pointing to bimodality. The authors then visually “chose” unimodal since there is strong overlap.

Experimental design

Project Design: on line 224 it states that more data was needed and on line 252-254 it says …as more data was taken into account….. What does this mean? It wasn’t mentioned in the methods or results and if more data could be taken into account to strengthen the result, why wasn’t it?

Validity of the findings

the title is Unimodal head width distribution….. even though the tests on lines 205-216 are largely inconclusive, one pointing to unimodality and two pointing to bimodality. The authors then visually “chose” unimodal since there is strong overlap.
In lines 220-222, uni and bimodality were equally supported between eel stages of eel.
My experience is that eel headshape distributions are a continuum, but are often heavily skewed in one direction or the other, varying widely between locations and habitats, even quite proximate ones. This paper picks a single eel population from the Schelde Estuary.
The abstract starts off with a categorical statement in line 26-27 and then concludes in line 31 that the distribution is unimodal and in lines 33-34 that it’s a continuum from narrow to broad, even though the statistics were a bit ambiguous. Lines 241 and 242 state equal support for uni and bimodality but then go on to say that uni is more “plausible”.
Section 3.3 relationship between HW and body condition, as defined by condition factor, was signif p<0.01 but with an r2 of .02, so only 2% of the variation in condition factor (body weight for length) is explained by HW – hardly convincing. Barry et al. and others have shown significantly lower lipid reserves in eels with boarder headwidths which is at odds with this finding.
The Migration speed section should be more clearly described and linked to the downstream migration of maturing silver eel (if that’s what it is). It is not clear to the more generalised reader. Especially when it is compared to home range movement of foraging immature yellow eels (i.e. Barry et al), which is a different thing.

Additional comments

I find it difficult to recommend this for publication without major revision. The context needs to be set to the Schelde Estuary and the title need to change. A more appropriate title might be something like: Distribution of headshape in European Eel in the Schelde Estuary,(with reference to body condition and migration speed?). The abstract needs to show the equal support for the two models and state that it is the authors visualisation that selected the unimodal form. Body condition increased with HW but only 2% of the variation was explained. Downstream migration speed was not influenced by head width.

Annotated reviews are not available for download in order to protect the identity of reviewers who chose to remain anonymous.

---

## Round 0.2 · Major Revisions

Though you have addressed the concerns from Reviewer 1, and some of mine, the manuscript still requires some revision.

Both reviewer 2 and myself have identified areas of text, particularly in the discussion, that don't seem well supported by your results and need to be amended. This is particularly clear in the swimming section. Reviewer 2 has provided a detailed list of these, which I would ask you to address.

Reviewer 1 ·

Basic reporting

no comment

Experimental design

no comment

Validity of the findings

no comment

Additional comments

It is my view that the current, second manuscript has been improved over the first and now fulfills the criteria and standards of the journal.

Reviewer 2 ·

Basic reporting

See below Note to Editor

Experimental design

See below Note to Editor

Validity of the findings

See below Note to Editor

Additional comments

I have reviewed this second version almost without reference to the first and I haven’t made any edits on the manuscript. I appreciate the suggested changes being taken into account.
1. Variation in headshape in eel is clear. This varies between eel sizes, locations, habitats and maybe even by sex and is linked with diet, niches, growth rates, lt/wt and other body conditions such as lipid levels. However, cause and effect is still strongly debated.

Much of the published literature, and the discussion by observers, fishers etc refers to them as an “either or” narrow or broad leading to this debate about bimodality.
Head shape is a continuum from an extreme pointy narrow head to an extreme blunt broad head with widely vary distributions along this continuum, sometimes leading to apparent bimodality within sites (such as Ide et al) or to heavily skewed distributions in one direction or the other. The combinations of these sites, and or the generalising in discussions, has perpetuated the view of “bimodality”. I’m saying this because I’m trying to decide what this current paper is actually trying to say.

In this paper, there is still a tendency to generalise to the whole population by stating that other authors show have shown bimodality (Torlitz, Ide & Kaifu). These examples are sampling eels from multiples of habitats and locations and then discuss a pooled bimodality. Your study is from one specific location within one specific river.

I reiterate my statement “that eel headshape distributions are a continuum, but are often heavily skewed in one direction or the other, varying widely between locations and habitats”. This does not mean there is no selective pressure in the Schelde. There are weak tails to your normal distributions which you ignore in line 216 and then you refer to them in lines 304-305. By analysing residuals, it is difficult to tell where your eels lie on the continuum.

Line 35, while correct, implies that there are no selective pressures on head shape, which in your case could be unidirectional selection pressure rather than disruptive pressure. This makes the point being made on line 98 confusing; yes biomodality can arise from disruptive selection, but equally unimodality can derive from unidirectional selection pressure. (And don’t discount the idea that eels can be quite choosey and selective in what they choose to eat – regardless of prey abundance)
I would suggest that if many of the references to bimodality were exchanged for “variations in head shape” the paper would read more accurately.
So for example, line 98 could read: From an evolutionary point of view, variations in headshape may arise from different selective pressures at many locations, or even disruptive pressure such as observed on the Frome (Cush…).
Line 306 repeats this bimodal statement, or absence of a clear bimodal distribution, but ignores that you were looking @ 1 niche, in 1 habitat type in 1 location. Line 308 states that this shows no selective pressure – on what basis? Or do you mean no disruptive selection pressure? You didn’t examine the diet in these fish and you don’t state where your eels lie on the continuum of the distribution of narrow to broad.
2. Following on from that, the discussion about body condition (Wt for length) supporting the lack of disruptive selection I don’t understand.
If there is a unimode, slightly skewed, there is likely to be some selective pressure towards that mode, but we don’t know where your eels lie on that distribution. Line 316 says that body condition increased with head width (body condition also increases with eel length). How does increasing K with headshape support a lack of disruptive selection when it such a weak relationship. And then the very low contribution (2%) actually supports it even more??
Written another way, body condition rises with bigger heads which means there is no selection, but because the relationship is so weak it really support this – that doesn’t make sense to me).
Related discussions on body weight, condition, energetics, and amount and size of prey in Lines 320-325 are quite speculative and don’t really move the debate further forward.

The statement in line 68 has not been supported: The consumption of different prey items not only leads to different head morphs, but can also influence habitat selection…..I suggest delete this – we don’t understand the cause and effects here.

Line 284-288 This doesn’t always apply to eel which can become quite specific to one food item, such as Asellus, in spite of an abundance of chironomid larvae, for example. And the size of eel can also influence the food types they prefer. So line 288 can conclude, but not always in eels.
Line 288-291 You haven’t related the diet of the individuals to their headshapes. You didn’t look at diet in your eels.

Line 319. You again apply cause and effect by stating that the eels with the broad heads occurred “as a result of feeding on hard prey,” – change this.
Line 320. They may be capable of eating a wide range of prey but I’m not sure they do.

3. 4.4 Swimming
There was no relationship with head width. So the rest of this paragraph is almost redundant. You are inferring something on the ocean migration from an upper estuarine observation of downstream movement, which is very different; flow rates, tidal flows, changes in salinity eel physiology and osmoregulation are all likely to influence swimming speed in the estuary, none of which are so relevant in the oceanic phase – we see similar differences in the speed of salmon smolts for example.

Line 344. This sentence is not supported and should be removed How does swimming speed (which wasn’t different between narrow and broad in the estuary) in the ocean explain how narrow heads have less wt for length – even though we know that narrow heads have more lipids (I acknowledge that this obs on lipids isn’t in the paper).

4. Conclusion
Line 344: Does it? You say dimorphism isn’t observed so why are you trying to explain it? You mean the absence of clear dimorphism in the Schelde estuary indicates a lack of evidence for disruptive selection, but doesn’t discount unidirectional pressures on variation in eels head shapes. Unimodal headshape in this location does not appear to subject to disruptive selection, but may have developed as a result of unidirectional selective pressure.

Line 348 Please remove this sentence (Bimodality is not a general phenomenon…..). Its not true.

The conclusions need to be reworded in the light of these comments (opportunism, one direction selective pressure, where are your eels on the distribution etc).

Line 353-354 also needs rewording.
Line 356 – delete “yet different morphs….. as this is not a conclusion of your study.

5. A few other bits
The catch location doesn’t say whether it was above or below the tidal weir. And a salinity range would be interesting if they were captured below the weir.

The Title: The title should say in the estuary of the Schelde – not “in a single river drainage”. This is exactly the issue under debate. They were from one location in one habitat in one river.

Methods: is there, or was there recently, any fishery in the estuary. Fishing pressure can influence the distribution of sizes and head shapes in eel stocks. Different growth rates, head shapes, dietary habits etc can all lead to selective harvest (though bait selection, mesh size etc) and a different distribution of head shape. This needs to be declared.


I still find it difficult to recommend this for publication without further major revision. I remain to be convinced that this paper as it is written has advanced our understanding much further.

---

## Round 0.3 · accepted · Accept

I have now re-examined the manuscript and am satisfied with the changes the authors have made in response to the reviewer's comments. I think the manuscript has benefitted from altering the interpretation of the results to acknowledge that only a single site was examined in the current study, and to provide a more broad discussion on selective pressures operating on this population (vs other populations) of eels.

My thanks to the reviewers for their time assessing the manuscript, and to the authors for making the necessary changes.

#